# Parenteral Nutrition in the Pediatric Oncologic Population: Are There Any Sex Differences?

**DOI:** 10.3390/nu15173822

**Published:** 2023-08-31

**Authors:** Laura De Nardi, Mariavittoria Sala, Federico Turoldo, Davide Zanon, Alessandra Maestro, Egidio Barbi, Barbara Faganel Kotnik, Natalia Maximova

**Affiliations:** 1Department of Medicine, Surgery and Health Sciences, University of Trieste, Piazzale Europa 1, 34127 Trieste, Italy; laura.denardi@burlo.trieste.it (L.D.N.); mavisala93@gmail.com (M.S.); turoldo97@gmail.com (F.T.); egidio.barbi@burlo.trieste.it (E.B.); 2Pharmacy and Clinical Pharmacology Department, Institute for Maternal and Child Health—IRCCS “Burlo Garofolo”, Via dell’Istria 65/1, 34137 Trieste, Italy; davide.zanon@burlo.trieste.it (D.Z.); alessandra.maestro@burlo.trieste.it (A.M.); 3Department of Pediatrics, Institute for Maternal and Child Health—IRCCS “Burlo Garofolo”, Via dell’Istria 65/1, 34137 Trieste, Italy; 4Department of Hematology and Oncology, University Children’s Hospital, 1000 Ljubljana, Slovenia; barbara.faganel@gmail.com

**Keywords:** total parenteral nutrition, pediatric patients, hematological malignancies, gender differences, sex disparities

## Abstract

Gender-based medicine is attracting increasing interest every day, but studies on pediatric populations are still limited. In this setting, sex differences among patients undergoing total parenteral nutrition (TPN) have not been previously reported. This study investigated the presence of sex differences in parenteral nutrition composition and outcomes among a cohort of pediatric patients admitted at the Oncohematology and Bone Marrow Transplant Unit of the Institute for Maternal and Child Health “Burlo Garofolo” of Trieste, Italy. For all 145 recruited patients (87 males, 58 females), the following data were collected: age, sex, volume and duration of TPN, macro- and micronutrient composition of TPN bags, electrolytic or blood gases imbalance, glycolipid alterations, liver damage during TPN, and the incidence of sepsis and thrombosis. The analysis showed that females required higher daily phosphate intake (*p* = 0.054) and essential amino acid supplementation (*p* = 0.07), while males had a higher incidence of hypertriglyceridemia (*p* < 0.05) and cholestasis. A higher incidence of sepsis was found in the non-transplanted male population (*p* < 0.05). No significant differences were appreciable in other analyzed variables. This study aims to create a basis for future gender-based nutritional recommendations in the pediatric field.

## 1. Introduction

Gender medicine in the pediatric field is only a very recent addition to clinical practice due to the limited availability of scientific evidence and ongoing research [1,2]. Differences in sex chromosomes, sex hormones, immune systems, and environmental exposures may contribute to sexual disparities in the incidence and severity of some diseases [3,4]. Females are reported to mount higher innate and adaptive immune responses than males, which could result, on the one hand, in faster clearance of pathogens and cancer cells and, on the other hand, contribute to increased susceptibility to inflammatory and autoimmune diseases. Unsurprisingly, males are more susceptible to infective diseases, develop a lower response to vaccines, and present a major cancer susceptibility and progression than females [5,6]. Some sex differences already begin in the uterus [7,8,9,10] and are detectable during pediatric age, as shown, for example, by the higher mortality of very preterm male infants who present with worse neurological and respiratory outcomes at follow-up [11]. Males are also at higher risk than females with respect to childhood cancer and its severity at any age [12,13], with more males affected by acute lymphoblastic leukemia (ALL) needing to undergo hematopoietic stem cell transplantation (HSCT) [14,15]. Furthermore, males are reported to develop Kawasaki disease more often in pediatric age and have worse cardiovascular outcomes in adulthood [16,17].

Total parenteral nutrition (TPN) is the administration of nutrients through central or peripheral venous access. It is a necessary therapeutic measure when a child is unable to feed adequately [18]. TPN regimens are built to provide adequate and balanced energy, macronutrients, and micronutrients to support growth and prevent deficiencies [19]. However, TPN can also bring some metabolic, thrombotic, or infective complications.

There is a significant lack of evidence about the impact of sex differences on TPN and its outcomes, especially in children, where the use of TPN is rare overall. TPN is used quite commonly in the pediatric oncologic setting, however, with frequent indications including chemotherapy-induced oral feeding intolerance, mucositis, and HSCT-related secondary intestinal failure [20], thus offering a good subject for investigation.

This study aims to explore sex differences in parenteral nutrition among a cohort of pediatric oncologic patients. The main objective is to analyze sex differences in nutritional needs by analyzing TPN bag volumes and composition during hospitalization. The study’s secondary goal is to explore whether any sex differences exist in the incidence of TPN-related complications. We aimed to find indicators that could lead to a better-personalized approach for pediatric patients.

## 2. Materials and Methods

### 2.1. Patient Population

This is a single-center retrospective cohort observational study conducted at the pediatric Oncohematology and Bone Marrow Transplant Unit of the Institute for Maternal and Child Health “Burlo Garofolo” of Trieste, Italy.

Eligible patients were pediatric patients aged between 0 and 18 years affected by hematological malignancies and non-malignancies who received TPN from January 2010 to December 2021.

The hematological conditions taken into account were acute lymphoblastic or myeloid leukemia (ALL, AML); solid tumors (neuroblastoma, Ewing sarcoma, Burkitt lymphoma, Hodgkin lymphoma, osteosarcoma, Wilms tumor, medulloblastoma, diffuse large B-cell lymphoma, myeloid sarcoma, mediastinal T-cell non-Hodgkin lymphoma, rhabdoid tumor, hepatoblastoma, and undifferentiated sarcoma); myelodysplastic syndromes (refractory cytopenia, juvenile myelomonocytic leukemia, Fanconi anemia, and aplastic anemia); and inborn errors of immunity (X-linked lymphoproliferative disease, CD40 ligand deficiency, and chronic granulomatous disease). The study population included the patients who underwent HSCT and those who underwent chemotherapy only. Indications for HSCT conformed to the American Society for Blood and Marrow Transplantation (ASBMT) task force recommendations for pediatric patients [21]. All patients were treated according to standard myeloablative protocols based on chemotherapy and radiation dosing, as previously described [22]. All the transplanted patients received tacrolimus as part of the immunosuppression regimen. High-dose post-transplant cyclophosphamide-based platforms were administered to all haploidentical recipients so that composite endpoints were comparable across conditioning intensity, donor type, and HLA match [23].

The study’s primary aim was to determine if any sex differences occur in the nutritional needs of oncological pediatric patients who undergo the administration of TPN. The secondary aim of the study was to explore sex differences in the incidence of TPN-related complications.

In our ward, TPN is provided by the hospital’s pharmacy, which creates personalized TPN bags for each patient according to their nutritional needs. These are evaluated and modified daily based on the laboratory analysis of blood samples. This provides standardization of care and assures every patient the appropriate intake of macro- and microelements.

All the patients enrolled in this study were analyzed individually. However, whenever the same patient received TPN in the setting of more than one hospitalization, the data about each hospitalization were treated like that of a different patient. All data were collected from clinical records and analyzed anonymously.

### 2.2. Variables

For each patient, the following variables were collected: demographic data, including age, sex, weight, and age at diagnosis; type of underlying disease; duration of TPN; volume and number of TPN bags; micronutrient composition in terms of the daily intake of calcium, phosphorus, and magnesium; and glucose and lipids composition. The following serum laboratory parameters were registered before and after the onset of TPN for each patient: glycemia, alanine aminotransferase (ALT), aspartate aminotransferase (AST), albumin, total proteins, and levels of calcium, magnesium, and phosphorus. The presence of an imbalance in serum levels of calcium (normal range 8.8–10.8 mg/dL), phosphorus (normal range 3–4.5 mg/dL), and magnesium (normal range 1.46–2.8 mg/dL) were registered both before and after TPN administration, along with the presence of hyperglycemia (given by blood glucose levels > 123 mg/dL) and the alteration of blood gases by metabolic acidosis or alkalosis.

Also, after TPN onset, the patients were tested for the presence of hypercholesterolemia (defined as low-density lipoprotein cholesterol—LDL-C > 150 mg/dL) and hypertriglyceridemia (TG >110 mg/dL).

All the serum referral values have been considered by age [24].

The amount of essential amino acids (EAA) and selective amino acids (SAA) administered in TPN for each patient was recorded. EAA supplementation was performed with an admixture containing L-Isoleucine, L-Leucine, Lysine base, L-Lysine Acetate, L-Phenylalanine, L-Threonine, L-Tryptophan, L-Valine, L-Histidine, and L-Methionine. It was preferred for patients with renal impairment since the lack of arginine in the mixture reduces the protein kidney overload. Conversely, it is reported to increase the risk of hyperammonemia, especially for those patients who present any liver impairment. Selective amino acid (SAA) formulae, in turn, which are rich in branched-chain amino acids, were used in patients with severe liver dysfunction that may lead to hepatic encephalopathy [25]. The SAA admixture used by our pharmacy contains L-Alanine, L-Arginine, L-Cysteine hydrochloride monohydrate, Glycine, L-Isoleucine, L-Histidine, L-Leucine, L-Lisine acetate, L-Proline, L-Serine, L-Threonine, L-Tryptophan, and L-Valine.

Two types of lipid admixtures were mainly used: Lipofundin S^®^ and Omegaven^®^. Lipofundin S^®^ is an admixture containing soy oil, while Omegaven^®^ is a compound made of fish oil, egg phospholipids, and dl-alpha-tocopherol, usually indicated for adult patients. As previously reported in the literature, specific lipid emulsions with varied fatty acid compositions, such as fish oil, may reduce complications associated with TPN lipid administration [26,27]. Hence, while the Lipofundin S^®^ was added to the PN bag of all patients to provide the daily lipid intake, the Omegaven^®^ was only added to the bags of those patients who presented severe liver damage, like PNAC or who needed additional immune system stimulation [28,29]. Over the course of years, the two compounds were progressively substituted by Smoflipid^®^, which contains refined soy oil, medium chain triglycerides, refined olive oil, and omega-3 enriched fish oil. This admixture could be nutritionally considered a combination of Lipofundin S^®^ and Omegaven^®^, thus making it easier to manage and reach a balanced lipid intake. However, in some selected cases of massive hypertriglyceridemia, Smoflipid^®^ has been replaced by Lipofundin S^®^ to benefit from the substitution of olive oil with faster clearance oils—in this case, soy oil [30].

In particular, the presence of liver injury, hyperammonemia, and cholestasis during the hospitalization was registered. Liver injury was defined as increased AST, ALT, or alkaline phosphatase > 2 times the upper limit of normal on two consecutive occasions; serum bilirubin > 2.5 mg/dL along with any elevations in AST, ALT, or alkaline phosphatase levels; or internationalized normal ratio (INR) > 1.5 with any elevations in AST, ALT, or alkaline phosphatase levels.

Finally, complications related to the use of a central venous catheter (CVC), such as thrombosis and sepsis, were recorded.

The presence of sex differences was analyzed among all these variables and then stratified through a multivariate logistic regression to ascertain if the results could be influenced by age, duration of parenteral nutrition, and whether a patient was undergoing HSCT or not. This was done to exclude any effect related to such other variables. Age, in this case, was considered a continuous variable. Finally, sex differences were examined both in the HSCT and non-HSCT populations to see if any sex differences occurred between transplanted and non-transplanted patients.

### 2.3. Ethical Approval

The Institutional Review Board of the IRCCS Burlo Garofolo (reference No. IRB RC 32/2023) approved the protocol, and the study was conducted following the Declaration of Helsinki. The data were collected according to the Authorization to Process Personal Data for Scientific Research Purposes (Authorization No. 9/2014) [31]. Parents signed informed consent at the first visit, agreeing that “clinical data may be used for clinical research purposes, epidemiology, the study of pathologies and training, to improve knowledge, care, and prevention”.

### 2.4. Statistical Analysis

All data were divided by the sex of the patient, and the two cohorts were compared. Data were statistically analyzed using R software version 4.2.2 (31 October 2022 ucrt) for Windows [32]. Descriptive statistics were used to analyze demographic data, clinical data, details on TPN usage, and complications using the “gtsummary” package [33]. Categorical data were presented as frequency and percentage, while continuous data were presented as the median and interquartile range (IQR). Tests defaulted were the Wilcoxon rank sum test for continuous variables, Pearson’s Chi-squared test without Yates’ correction for categorical variables with all expected cell counts ≥ 5, and Fisher’s exact test for categorical variables with any expected cell count < 5. Incomplete observations were removed from the analysis.

The relationship between predictors and dependent categorical variables was investigated through a multivariate logistic regression with the help of the “epiDisplay” package [34]. The relationships between predictors and dependent continuous variables were investigated through multivariate linear regression. Incomplete observations were removed from the analysis. Multivariate regression was adjusted according to age, duration of TPN, and history of HSCT. Furthermore, regarding some relevant complications, additional variables were included in the adjustment: hyperammonemia was also adjusted according to the supplement of EAA; cholestasis and hypertriglyceridemia were also adjusted according to the supplement of glucose and lipids. All analyses were performed separately in the whole cohort, in the cohort of patients who underwent HSCT, and in the cohort of patients who did not undergo HSCT.

## 3. Results

Two hundred sixty-seven patients were admitted to the Institute for Maternal and Child Health “Burlo Garofolo” of Trieste, Italy, for malignant and hematological disease between 2010 and 2021. One hundred twenty-two patients were excluded from the analysis because they did not fulfill the inclusion criteria. One hundred forty-five patients (87 males, 58 females) who met the inclusion criteria were enrolled. Since some of the patients received TPN more than once in their life, for statistical convenience criteria, they were considered as different patients in each TPN period, turning the sample size into a population of 273 patients, of which 164 (60%) were males, and 109 (40%) were females. Among these patients, 202 received HSCT (117 males and 85 females), and 71 received high-dose chemotherapy (47 males, 24 females). Patients enrolled in the study fell into five main disease groups: acute lymphoblastic leukemia (42%), solid tumors (32%), acute myeloid leukemia (12%), myelodysplastic syndromes (11%), and inborn immunity errors (3%).

Population descriptive analysis is shown in Table 1. The mean age in the male group was 9 years old (interquartile age range: 4–15 years old), while in the female group, it was 12 years old (interquartile age range: 8–14 years old). The mean duration of the TPN period was 11 days for males and 12 days for females, with no statistically significant difference. 126 (77%) male patients and 79 (72%) females underwent HSCT. The types of HSCT performed were autologous (26% M, 20% F), allogenic (17%M, 11% F), matched unrelated donor (MUD) (29% M, 33% F), and haploidentical (4.9% M, 8.3% F). The average weight was 32 kg (interquartile range: 18, 54) for males and 41 (interquartile range: 24, 51) kg for females. Unfortunately, weight data were not available for part of the population. No significant sex difference was found in the prevalence of electrolyte imbalance before TPN began, even if data were not available for part of the population. The most common electrolyte imbalance was hypocalcemia, which was present in 61 patients.

The descriptive analysis of the characteristics of the population and the composition of TPN bags separately for the HSCT cohort and the not-HSCT cohort is available in the Appendix A.

All the collected variables were analyzed to evaluate any sex differences. In this first analysis, females received greater daily phosphate (*p* = 0.057) supplementation than males in their TPN bag, while males received greater calcium supplementation (*p* = 0.033). However, only the females’ greater daily phosphate intake was confirmed in a second multivariate logistic regression analysis: linear multivariate regression of continuous variables with respect to the composition of TPN bags in the total cohort is shown in the Appendix A. Daily calcium intake in the multivariate analysis, however, was found to be associated with longer TPN periods.

Furthermore, a slight difference in the supplementation of EAA was found, which was more common in females *(p* = 0.072). In the multivariate analysis, such a difference was confirmed in the total population (*p* = 0.068) and depended on age. No statistically significant sex differences were appreciated in the multivariate analysis of calcium, magnesium, glucose, lipids, SAA supplementation, and total volume administered (*p* > 0.05).

The incidence of TPN complications is shown in Table 2. The incidence of complications for the HSCT cohort and the cohort that did not undergo HSCT is available separately in the Appendix A.

The most common complication was liver injury, which was present in 194 patients. On the other hand, hypercalcemia, hypermagnesemia, and hematic pH alterations were rare after the beginning of TPN.

Males presented with more hypertriglyceridemia (*p* = 0.042) and cholestasis (*p* = 0.07) after TPN began. These results have also been confirmed by the second stratified multivariate analysis (Appendix A). Only the most relevant results of the multivariate logistic regression shown in Appendix A are reported in Table 3.

The multivariate regression confirmed a higher incidence of cholestasis and hypertriglyceridemia in males. Analysis was also adjusted according to glucose and lipids intake in order to eliminate confounding variables. Cholestasis (F vs. M OR: 0.45 95% CI: 0.22, 0.95) was also strongly associated with undergoing HSCT (OR 8.43, 95% CI: 1.96, 36.21) and also with lipids and glucose supplements. A greater male hypertriglyceridemia incidence was also confirmed (F vs. M OR 0.45 95% CI: 0.23, 0.88).

A slightly higher incidence of hyperphosphatemia after TPN in males was also observed *(p* = 0.072), but only in the HSCT group. Also, a significant difference in the incidence of metabolic alkalosis was found *(p* = 0.013), but the size of this group was small, and this limited further analysis.

Furthermore, a greater risk of sepsis was found in non-HSCT males (*p* = 0.033), and the difference was also present after the adjustments (*p* = 0.065) but was not confirmed in the HSCT group or in the general population. Also, the multivariate analysis performed on the total cohort showed that HSCT was a protective factor for sepsis (OR 0.39, 95% CI: 0.18, 0.82).

## 4. Discussion

Our investigation is the first study dealing with sex differences in parenteral nutrition among a cohort of oncologic pediatric patients.

In our study, a greater phosphate female daily intake was detected, which positively correlated with older age (*p* = 0.073). In accordance with the data in the literature, no other significant differences in daily micronutrient intake between males and females were detected [35]. Phosphorus is one of the most abundant elements in the human body and is mainly found in complexes with oxygen in the form of phosphate. It is found in bones and teeth, but also in soft tissue, in the intracellular compartment as an essential component of several organic compounds, including nucleic acids and cell membrane phospholipids, and also involved in aerobic and anaerobic energy metabolism [36].

Hematic phosphate levels can vary according to renal function, endocrine factors, hematic pH, and other electrolyte levels, and are also sensible to malnutrition and refeeding syndrome. In fact, alteration in phosphate levels is a common finding in TPN patients, and phosphate supplementation is essential to prevent hypophosphatemia [37,38].

It has been shown that the female sex could represent a risk factor for hypophosphatemia in postoperative patients with TPN [39]. This could suggest that female patients require higher phosphate supplementation to balance their physiological functions than males and that this could also be true in the pediatric age.

Also, levels of hematic phosphate and phosphate absorption are regulated by Vitamin D, whose metabolism and activity are well known to vary according to sex [40]. Furthermore, sex differences were found in acute phosphate homeostasis. Specifically, females mobilize and excrete more endogenous sources of calcium and phosphate in response to oral phosphate when compared to males [41]. For these reasons, it seems reasonable that sex can influence phosphate intake, even if this claim should be further confirmed [42].

In addition, female patients showed a greater intake of EAA when compared to males. This sex difference was also confirmed in the HSCT subgroup. Essential amino acids supplementation is used mainly in renal impairment since the lack of arginine in the mixture reduces protein kidney overload. Conversely, this can increase the risk of hyperammonemia [43,44,45]. However, no sex differences in the incidence of hyperammonemia were found, not even after adjusting for EAA supplementation in the multivariate analysis (Appendix A).

Turning to lipid metabolism, males showed a higher incidence of hypertriglyceridemia than females (*p* = 0.042), and this difference is preserved within the HSCT population. Data on cholesterol levels showed the same trend, although without statistical significance. Hypertriglyceridemia is one of the most frequent TPN-related complications secondary to the TPN lipid administration and is a common finding in patients undergoing HSCT [46]. HSCT is associated with a higher incidence of hypertriglyceridemia due to prolonged rest periods, hormonal dysregulation, and the side effects of medicines, particularly the glucocorticoids, Janus kinase, and calcineurin inhibitors that are used to prevent and treat GVHD [47,48,49,50]. In such cohorts of patients, the male sex has sometimes been related to hypertriglyceridemia, but there is no uniform consensus. The variability of results is most likely due to the heterogeneity of conditioning regimens, immunosuppressive therapies, and nutritional support—since not all patients in the analyzed cohorts were fed exclusively through TPN. Although the percentage of hypercholesterolemia was slightly higher in males, the only statistically significant association found in the analysis was with older age.

Since one of the pathophysiological mechanisms causing hypercholesterolemia and hypertriglyceridemia is drug-induced cholestasis, with the impaired bile flow leading to an increase in plasma lipid concentration, it is not surprising that cholestasis has also been found to have a significant incidence among male patients. Parenteral nutrition-associated cholestasis (PNAC) affects 20–40% of pediatric patients undergoing TPN and is closely related to its duration [51,52]. The results of our study are coherent with what is reported in the literature, where a higher incidence of cholestasis in male patients is described. An investigation on newborns who are surgical patients showed a predisposition of the male sex to PNAC [53]. Another showed that it was significantly more frequent in males when they received TPN for an extended period [54]. One explanation for this male disadvantage in newborns is lower duodenal bile salt concentration [55].

Although the correlation between male sex and PNAC has not been explained, it could be an exciting starting point for future research. The available evidence seems to suggest that the type of lipids used in TPN is an important factor in the incidence of PNAC [26]. In our study, as already mentioned, the validity of information about the type of lipids administered was compromised by a change in the protocol during the study period. As a matter of fact, the multivariate logistic regression confirmed the strong association between PNAC and the number of daily lipids and glucose administered, but the influence of sex was also confirmed after taking these variables into account. The multivariate logistic regression also showed an association between PNAC and undergoing HSCT, which is probably influenced by the duration of the TPN and the use of hepatotoxic drugs in this cohort.

Sepsis has also been described as being linked with PNAC since coexisting systemic or intraluminal infection and intestinal stasis with prolonged starvation play significant roles in the development of PNAC.

Although several studies have explored this relationship, there is still no consensus on whether they are related through a causative link or if sepsis is only secondary to PNAC [53,56]. In this study, a greater incidence of sepsis was found in non-HSCT males (*p* = 0.033), in accordance with the available literature that considers males more susceptible to infectious diseases [5,6]. However, this sex difference disappears in the HSCT group (*p* = 0.7). It was surprising to find that HSCT patients, who are supposed to be immunosuppressed, have less risk of sepsis than non-HSCT ones (OR 0.39, 95% CI: 0.18, 0.82), as seen in Appendix A. This finding could be explained by the likelihood that HSCT patients undergo stricter measures and protocols to prevent infection.

A further potential explanation for the disappearance of the sex difference in the HSCT group could be found in the theory that the HSCT and its previous conditioning regimen could flatten the hormonal differences between males and females for a certain period, reducing the influence of sex on transplant-related outcomes, as some studies suggest [57]. According to this thesis, it would be possible to witness an equalization in the incidence of clinical situations that would otherwise be influenced by sex. Indeed, such a hypothesis should be tested in a larger sample size.

Otherwise, no sex differences were found in the incidence of catheter-related thrombosis.

In conclusion, our study seems to confirm some gender differences in TPN complications, especially regarding cholestasis and hypertriglyceridemia, which are more frequent in males. On the other hand, TPN micro-nutrient needs were mostly similar in both sexes, with the few exceptions being EAA and phosphate. Other associations were found but were less consistent across stratified analyses.

The current study presents some limitations related to the fact that some data were missing for some patients, who were then excluded from the analysis so as not to affect the results negatively. Also, since each patient was analyzed more than once, any possible differences between periods of nutritional treatment were not considered.

Secondly, we did not perform a stratified analysis for primary diseases and treatment protocols. Further study could consider the influence of drugs to provide more robust conclusions. Finally, the changes implemented in the type of lipid used over the years could have influenced the results on some complications such as cholestasis since it is known that lipid emulsions based on fish oil with a high content of long-chain polyunsaturated fatty acids ω-3 appear effective both in decreasing intrahepatic inflammation and in improving biliary flow [27].

However, this study also has its strengths. It is the first report in the literature to explore sex differences in the use and outcomes related to parenteral nutrition. The only study in the literature deals with a difference in metabolic amino acid expression by sex in newborns, influencing the referral values of newborn screening [58]. Finally, the presence of an internal pharmacy at the hospital ensures both the standardization of TPN bag components as well as the daily modification of their proportions to provide the best supplementation for the daily needs of each patient.

## 5. Conclusions

In conclusion, this study has shown the presence of some gender differences in TPN-related complications and micronutrient needs within a cohort of oncologic pediatric patients. In particular, it has demonstrated a major need for daily phosphate intake and essential amino acids in female patients. At the same time, it has shown a significant incidence in males of cholestasis and hypertriglyceridemia related to TPN. Further studies are needed to confirm these results in a larger cohort of patients to ensure a more personalized approach for pediatric patients.

## Figures and Tables

**Table 1 nutrients-15-03822-t001:** Descriptive analysis of the population and the composition of TPN bags.

Characteristic	M, *N* = 164 ^1^	F, *N* = 109 ^1^	*p*-Value ^2^
Age (years)	9.0 (4.0, 15.0)	12.0 (8.0, 14.0)	0.11
Age group			0.5
>12 years	59 (36%)	44 (40%)	
0–12 years	105 (64%)	65 (60%)	
TPN duration (days)	11 (6, 19)	12 (6, 21)	0.5
HSCT			0.4
0—No	38 (23%)	30 (28%)	
1—Yes	126 (77%)	79 (72%)	
HSCT type			0.3
0—No HSTC	38 (23%)	30 (28%)	
1—Autologous	42 (26%)	22 (20%)	0.3
2—Allogenic	28 (17%)	12 (11%)	0.2
3—MUD	47 (29%)	36 (33%)	0.4
4—Haploidentical	8 (4.9%)	9 (8.3%)	0.3
Unknown	1	0	
Weight (kg)	32 (18, 54)	41 (24, 51)	0.6
Unknown	19	14	
Total number of TPN bags	11 (6, 19)	13 (6, 20)	0.6
Vol/die (mL)	1800 (1184, 2400)	1959 (1320, 2446)	0.4
Magnesium/die (mg)	200 (100, 316)	202 (134, 311)	0.5
(mmol)	8.23 (4.11, 13.00)	8.31 (5.51, 12.79)	
Phosphorus/die (mg)	300 (166, 502)	400 (200, 571)	0.057
(mmol)	9.69 (5.36, 16.21)	12.91 (6.46, 18.44)	
Calcium/die (mg)	335 (170, 600)	210 (94, 500)	0.033
(mmol)	8.35 (4.24, 14.97)	5.24 (2.34, 12.47)	
Glucose/die (g)	182 (118, 266)	211 (127, 254)	0.3
Lipids (g)	17 (10, 32)	23 (9, 33)	0.3
Hypocalcemia before TPN			>0.9
0—No	94 (72%)	62 (72%)	
1—Yes	37 (28%)	24 (28%)	
Unknown	33	23	
Hypomagnesemia before TPN			0.8
0—No	114 (87%)	74 (86%)	
1—Yes	17 (13%)	12 (14%)	
Unknown	33	23	
Hypophosphatemia before TPN			0.2
0—No	112 (85%)	67 (78%)	
1—Yes	19 (15%)	19 (22%)	
Unknown	33	23	
SAA supplementation			0.5
0—No	92 (56%)	57 (52%)	
1—Yes	72 (44%)	52 (48%)	
EAA supplementation			0.072
0—No	151 (92%)	92 (85%)	
1—Si	13 (7.9%)	16 (15%)	
Unknown	0	1	

TPN—total parenteral nutrition; HSCT—hematopoietic stem cell transplantation; MUD—matched unrelated donor; SAA—selective amino acids; EAA—essential amino acids. ^1^ Median (IQR); *n* (%), ^2^ Wilcoxon rank sum test; Pearson’s Chi-squared test, Fisher’s exact test.

**Table 2 nutrients-15-03822-t002:** Incidence of complications of TPN in the total cohort.

Characteristic	Male (*n* = 164) ^1^	Female (*n* = 109) ^1^	*p*-Value ^2^
Hypercalcemia after TPN			0.4
0—No	129 (100%)	83 (99%)	
1—Yes	0 (0%)	1 (1.2%)	
Unknown	35	25	
Hypermagnesemia after TPN			>0.9
0—No	127 (98%)	83 (99%)	
1—Yes	2 (1.6%)	1 (1.2%)	
Unknown	35	25	
Hyperphosphatemia after TPN			0.3
0—No	75 (59%)	55 (65%)	
1—Yes	53 (41%)	29 (35%)	
Unknown	36	25	
Increased transaminases			0.3
0—No	38 (28%)	31 (35%)	
1—Yes	97 (72%)	57 (65%)	
Unknown	29	21	
Cholestasis			0.07
0—No	128 (79%)	95 (87%)	
1—Yes	35 (21%)	14 (13%)	
Unknown	1	0	
Hyperammonemia			0.2
0—No	150 (91%)	103 (95%)	
1—Yes	14 (8.5%)	5 (4.6%)	
Unknown	0	1	
Hypercholesterolemia			0.3
0—No	135 (82%)	95 (87%)	
1—Yes	29 (18%)	14 (13%)	
Hypertriglyceridemia			0.042
0—No	125 (76%)	94 (86%)	
1—Yes	39 (24%)	15 (14%)	
Hyperglycemia			0.8
0—No	132 (80%)	89 (82%)	
1—Yes	32 (20%)	20 (18%)	
Liver injury			0.7
0—No	49 (30%)	30 (28%)	
1—Yes	115 (70%)	79 (72%)	
Metabolic acidosis			0.2
0—No	157 (96%)	108 (99%)	
1—Yes	7 (4.3%)	1 (0.9%)	
Metabolic alkalosis			0.013
0—No	155 (95%)	109 (100%)	
1—Yes	9 (5.5%)	0 (0%)	
Respiratory alkalosis			0.4
0—No	164 (100%)	108 (99%)	
1—Yes	0 (0%)	1 (0.9%)	
Sepsis			0.4
0—No	135 (82%)	94 (86%)	
1—Yes	29 (18%)	15 (14%)	
CVC thrombosis			0.7
0—No	140 (85%)	91 (83%)	
1—Yes	24 (15%)	18 (17%)	

TPN—total parenteral nutrition; CVC—central venous catheter. ^1^ Median (IQR); *n* (%), ^2^ Wilcoxon rank sum test; Pearson’s Chi-squared test, Fisher’s exact test.

**Table 3 nutrients-15-03822-t003:** Multivariate logistic regression (categorical variables).

Characteristic 1 vs. 0		OR (95% CI)	*p*-Value ^1^
Cholestasis			
	Gender: F vs. M	0.45 (0.22, 0.95)	0.035
	Age *	1.06 (0.99, 1.13)	0.085
	TPN duration *	1.02 (1, 1.04)	0.078
	HSCT: 1 vs. 0	9.09 (2.08, 39.8)	0.003
	Lipide/die (stand. ^2^) *	2.66 (1.26 ,5.59)	0.01
	Glucose/die (stand. ^2^) *	0.32 (0.12, 0.89)	0.028
Hypertriglyceridemia			
	Gender: F vs. M	0.45 (0.23, 0.88)	0.019
	Age *	1.05 (0.99, 1.12)	0.079
	TPN duration *	1.02 (1, 1.04)	0.075
	HSCT: 1 vs. 0	0.72 (0.35, 1.49)	0.376
	Lipide/die (stand. ^2^) *	1.38 (0.84, 2.26)	0.198
	Glucose/die (stand. ^2^) *	0.86 (0.46, 1.6)	0.635
Sepsis (not-HSCT)			
	Gender: F vs. M	0.26 (0.06, 1.09)	0.065
	Age *	1.11 (0.99, 1.24)	0.07
	TPN duration *	1.04 (0.96, 1.13)	0.284
Hyperphosphatemia (HSCT)			
	Gender: F vs. M	0.51 (0.25, 1.06)	0.072
	Age *	0.92 (0.87, 0.98)	0.012
	TPN duration *	0.99 (0.96, 1.01)	0.306

OR—odds ratio; TPN—total parenteral nutrition; HSCT—hematopoietic stem cell transplantation; EAA—essential amino acids. * continuous variable. ^1^ Wald’s test. ^2^ Lipide/die and Glucose/die was standardized to obtain average = 0 and standard deviation = 1.

## Data Availability

The data that support the findings of this study are available from the corresponding author upon request.

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
