# Peer review of "Parenteral Nutrition in the Pediatric Oncologic Population: Are There Any Sex Differences?"

_nutrients, 2023, doi:10.3390/nu15173822_

Round 1

Reviewer 1 Report

The relevance of the study is questionable and has little clinical importance, since every Parenteral Nutrition (PN) prescription for patients with this degree of underlying disease must be individualized. The differences between the genders found were cholestasis and higher hypertriglyceridemia in males. Today it is known that one of the major factors involving these complications is the type of lipid used, and this was not mentioned in the study. As for the greater supply of magnesium in females, I could not understand why, since hypomagnesemia was not significantly different between the two sexes. As for Essential Amino Acids, it is assumed that their indication would be associated with renal failure and this was not presented either.

In conclusion, I think that the study does not present elements that justify special care for those who prescribe PN for these patients, and an analysis of some additional aspects such as those mentioned would add more objective information.

I noticed that in the Abstract the abbreviation HSCT was not presented in full, and I think it should be added.

Author Response

We thank the reviewers for the comments and are glad to provide a manuscript revision. We modified the text according to your suggestions in order to answer the issues raised by the reviewers (corrections are highlighted in red in the marked copy).

1.1- The relevance of the study is questionable and has little clinical importance since every Parenteral Nutrition (PN) prescription for patients with this degree of underlying disease must be individualized.

It is certainly true that every TPN prescription for oncologic patients must be individualized. Hence it is extremely difficult to draw conclusions that can be generalized on a large scale. However, we think the relevance of this study resides in the availability of a large sample size to whom TPN is routinely administered in the oncologic setting. This allowed us to explore sex differences in TPN composition and complications, performing a gender analysis which is something unexplored in literature, especially in the pediatric field.

We think that the implications of such results, which clearly need to be confirmed, could improve patients’ care towards an even more individualized approach that also considers sex differences.

1.2- The differences between the genders found were cholestasis and higher hypertriglyceridemia in males. Today it is known that one of the major factors involving these complications is the type of lipid used, and this was not mentioned in the study.

The reviewer is right; we added in the text the type of lipid used along with the relative references on the topic. Since this retrospective cohort observational study collected patients admitted for over 10 years at our Oncohematology and Bone Marrow Transplant Unit, some changes in the clinical practice have been performed over the years. Two lipid admixture types were first used: Lipofundin S® and Omegaven®. Lipofundin S® is an admixture containing soy oil, while Omegaven®  is a compound made of fish oil, egg phospholipids, and dl-alpha-tocopherol, usually indicated for adult patients. As previously reported in the literature, specific lipid emulsions with varied fatty acid composition, such as fish oil, may reduce complications associated with TPN lipid administration (1,2). Hence, while the Lipofundin S® was added to the PN bag of all patients to provide the daily lipid intake, the Omegaven® was only added to the bags of those patients who presented severe liver damage, like PNAC or who needed additional immune system stimulation (3,4). During the course of the years, the two compounds were progressively substituted by Smoflipid®, which contains refined soy oil, medium chain triglycerides, refined olive oil, and omega-3 enriched fish oil. This admixture could be nutritionally considered a combination of Lipofundin S® and Omegaven®, thus making it easier to manage and reach a balanced lipid intake. However, in some selected cases of massive hypertriglyceridemia, Smoflipid® has been replaced by Lipofundin S® to benefit from substituting olive oil with faster clearance oils – in this case, soy oil (5). The changes in the type of lipid used over the years could have mildly influenced the results on cholestasis, which was added in the text among the study's limitations.

  1. Guthrie G, Burrin D. Impact of Parenteral Lipid Emulsion Components on Cholestatic Liver Disease in Neonates. Nutrients. 2021 Feb 4;13(2):508.
  2. Orso G, Mandato C, Veropalumbo C, Cecchi N, Garzi A, Vajro P. Pediatric parenteral nutrition-associated liver disease and cholestasis: Novel advances in pathomechanisms-based prevention and treatment. Dig Liver Dis. 2016 Mar;48(3):215–22.
  3. Diamond IR, Sterescu A, Pencharz PB, Kim JH, Wales PW. Changing the Paradigm: Omegaven for the Treatment of Liver Failure in Pediatric Short Bowel Syndrome. J Pediatr Gastroenterol Nutr. 2009 Feb;48(2):209–15.
  4. Allam-Ndoul B, Guénard F, Barbier O, Vohl MC. A Study of the Differential Effects of Eicosapentaenoic Acid (EPA) and Docosahexaenoic Acid (DHA) on Gene Expression Profiles of Stimulated Thp-1 Macrophages. Nutrients. 2017 Apr 25;9(5):424.
  5. Mateu-de Antonio J, Florit-Sureda M. New Strategy to Reduce Hypertriglyceridemia During Parenteral Nutrition While Maintaining Energy Intake. J Parenter Enter Nutr. 2016 Jul;40(5):705–12.

1.3- As for the greater supply of magnesium in females, I could not understand why, since hypomagnesemia was not significantly different between the two sexes.

In the revised version of the paper that we attached, after an additional round of analysis performed to meet also other reviewers’ suggestions, some changes emerged relative to the supply of electrolytes due to an edit in the dataset and statistical procedure. In particular, the difference in magnesium supply wasn’t found to be statistically significant, and thus that the previous result could be an inconsistency. However, a slightly significant difference in the phosphate supply emerged relative to the total population (p=0.054), even if the incidence of hypophosphatemia was similar between the sexes.   

 The fact that no significant differences in hypophosphatemia before the start of TPN have been detected does not automatically exclude the possibility that females have a higher phosphate need during the course of TPN administration. We do not consider such data contradictory because electrolyte alterations before the beginning of TPN and basal electrolyte needs during TPN are not necessarily linked to one another. In fact, we estimated that any electrolyte deficiency before TPN administration could be mostly influenced by the state of undernutrition preceding TPN itself and by the type of treatment, while the daily electrolyte supplementation could be more directly linked to the basal needs of the patient in a “stable” nutritional condition since TPN supposedly treats and prevents the aforementioned malnutrition.

Furthermore, individual electrolyte intake was adjusted based on individual electrolyte blood levels, and therefore the supplementation could be provided even if the patient didn’t meet the criteria for hypophosphatemia per se. This can lead to a sex difference in the daily intake, even if there is no sex difference in the hypophosphatemia rate.

1.4- As for Essential Amino Acids, it is assumed that their indication would be associated with renal failure and this was not presented either.

Thank you for the comment. As mentioned in the text (section 2.2 “Variables”), Essential Amino Acids (EAA) supplementation was mainly used for patients with renal impairment since the lack of arginine in the mixture reduces the protein kidney overload. Conversely, this can increase the risk of hyperammonemia, especially for those patients who present any liver impairment. In this paper, we did not focus on the presence of renal impairment since data on creatinine levels were not available for all patients. No sex differences were found in the incidence of hyperammonemia adjusted for AAE at the multivariate logistic regression (Supplements Table S3).

1.5- In conclusion, I think that the study does not present elements that justify special care for those who prescribe PN for these patients, and an analysis of some additional aspects such as those mentioned would add more objective information.

We added in the text the issues suggested by the reviewer. We think that the implications of such results could improve patients’ care towards an even more individualized approach that also considers sex differences. Indeed, while the prescription of the TPN formula may not be affected by this study since, as we mentioned, each bag composition is usually tailored to the patient’s current needs, a further comprehension of any sex difference in PN needs and complications may help improve the whole therapeutic path. This study, which is one of the first studies in such a setting, aims at raising awareness about sex differences in the pediatric oncological field, underlines higher TPN-related complications risk in sex subgroups, and could serve as a base for further studies about the pathophysiology of such differences.

1.6- I noticed that in the Abstract the abbreviation HSCT was not presented in full, and I think it should be added.

We modified the text.

Reviewer 2 Report

This is a very interesting retrospective analysis of sex-differencies of TPN compostion and complicatiy patoentns in pediatric oncology patients. The study is well designed, written and presented.

However I have some questions to authors

1. Did you analyse macronutrients provision in TPN, e.g. proteins, lipids, carbohydrates and energy

2. Regrading PNAC - what kind of lipids did you use in TPN, where they this same for all patients

I would suggest to include conclusion in the abstract section

Author Response

We thank the reviewers for the comments and are glad to provide a manuscript revision. We modified the text according to your suggestions in order to answer the issues raised by the reviewers (corrections are highlighted in red in the marked copy).

This is a very interesting retrospective analysis of sex-differencies of TPN compostion and complicatiy patoentns in pediatric oncology patients. The study is well designed, written and presented.

However I have some questions to authors

  • Did you analyse macronutrients provision in TPN, e.g. proteins, lipids, carbohydrates and energy

In the first analysis run, lipids and glucose intake data were collected but not included in the final analysis because we focused more on electrolyte imbalance. However, the analysis of macronutrient intake and basal necessities could be an interesting point for further study. In the revised version of the paper that we attached, after an additional round of analysis to meet also other reviewers’ suggestions, we also included data on lipids and glucose. No significant differences in the supply of these macronutrients were found, and the difference in TPN-related cholestasis and hypertriglyceridemia was also confirmed after considering glucose and lipid supply (Table S3, Supplements).

  • Regrading PNAC - what kind of lipids did you use in TPN, where they this same for all patients.

Two lipid admixture types were first used: Lipofundin S® and Omegaven®. Lipofundin S® is an admixture containing soy oil, while Omegaven®  is a compound made of fish oil, egg phospholipids, and dl-alpha-tocopherol, usually indicated for adult patients. As previously reported in the literature, specific lipid emulsions with varied fatty acid composition, such as fish oil, may reduce complications associated with TPN lipid administration (1,2). Hence, while the Lipofundin S® was added to the PN bag of all patients to provide the daily lipid intake, the Omegaven® was only added to the bags of those patients who presented severe liver damage, like PNAC or who needed additional immune system stimulation (3,4). During the course of the years, the two compounds were progressively substituted by Smoflipid®, which contains refined soy oil, medium chain triglycerides, refined olive oil and omega-3 enriched fish oil. This admixture could be nutritionally considered a combination of Lipofundin S® and Omegaven®, thus making it easier to manage and reach a balanced lipid intake. However, in some selected cases of massive hypertriglyceridemia, Smoflipid® has been replaced by Lipofundin S® to benefit from substituting olive oil with faster clearance oils – in this case, soy oil (5). The changes in the type of lipid used over the years could have mildly influenced the results on cholestasis, which was added in text among the study's limitations.

  1. Guthrie G, Burrin D. Impact of Parenteral Lipid Emulsion Components on Cholestatic Liver Disease in Neonates. Nutrients. 2021 Feb 4;13(2):508.
  2. Orso G, Mandato C, Veropalumbo C, Cecchi N, Garzi A, Vajro P. Pediatric parenteral nutrition-associated liver disease and cholestasis: Novel advances in pathomechanisms-based prevention and treatment. Dig Liver Dis. 2016 Mar;48(3):215–22.
  3. Diamond IR, Sterescu A, Pencharz PB, Kim JH, Wales PW. Changing the Paradigm: Omegaven for the Treatment of Liver Failure in Pediatric Short Bowel Syndrome. J Pediatr Gastroenterol Nutr. 2009 Feb;48(2):209–15.
  4. Allam-Ndoul B, Guénard F, Barbier O, Vohl MC. A Study of the Differential Effects of Eicosapentaenoic Acid (EPA) and Docosahexaenoic Acid (DHA) on Gene Expression Profiles of Stimulated Thp-1 Macrophages. Nutrients. 2017 Apr 25;9(5):424.
  5. Mateu-de Antonio J, Florit-Sureda M. New Strategy to Reduce Hypertriglyceridemia During Parenteral Nutrition While Maintaining Energy Intake. J Parenter Enter Nutr. 2016 Jul;40(5):705–12.
  • I would suggest to include conclusion in the abstract section.

We added a conclusion in the abstract section, as suggested.

Reviewer 3 Report

The paper contains many statistical analyses. The extensive tables in the text can be replaced by describing the lack of statistical significance in most of the data. The data can be shown in the supplementary materials. 

The reported intravenous supplementation expressed in mg/dL should also be given in SI units. 

It is unclear what giving Essential Amino Acids and Selective Amino Acids means. What amino acid solutions were administered? Two different preparations? It is not clear what preparations of lipid emulsions were used. What is the relationship with the occurrence of cholestasis depending on the treatment, including lipid and glucose supply?

The large age range of the subjects, 0-18 years, justifies providing information on the doses used in parenteral nutrition. 

Merely providing information about hyper or hypo - the result of the study of minerals, for example, is insufficient. 

It seems to me that the whole construction of the tables is difficult to interpret.

The daily volume of TPN, and time of application of TPN should not be evaluated without reference to the weight and age of the subjects.

As I understand it, since the nutritional mixtures were prepared in the hospital pharmacy individually for each patient, the requirement for personalized approval for pediatric patients was met, yet such a conclusion was made in the conclusions.

Since a single patient could be analyzed multiple times, were there differences in the studies between periods of nutritional treatment?

There are a lot of caveats that should be clarified if the study could be useful in clinical practice.

Author Response

We thank the reviewers for the comments and are glad to provide a manuscript revision. We modified the text according to your suggestions in order to answer the issues raised by the reviewers (corrections are highlighted in red in the marked copy).

3.1- The paper contains many statistical analyses. The extensive tables in the text can be replaced by describing the lack of statistical significance in most of the data. The data can be shown in the supplementary materials. 

Thank you for your suggestion. We feel is very appropriate. In the revised version that we attached, we only included 3 tables.  Descriptive analyses are kept in Table 1 and Table 2 to give the reader an idea of the characteristics of the study population. Regarding the following multivariate analysis, statistically insignificant results have been moved to the supplementary section, and only the most relevant results are reported in Table 3.

3.2- The reported intravenous supplementation expressed in mg/dL should also be given in SI units. 

In Table 1 and Table 2, we reported the average quantity of nutrients administered and not the concentration of the solution: since the prescription in our ward is given in mg, we feel that this was the appropriate unit.

3.3- It is unclear what giving Essential Amino Acids and Selective Amino Acids means. What amino acid solutions were administered? Two different preparations?

As mentioned in the text (Section 2.2 “Variables”), EAA supplementation was performed with an admixture containing L-Isoleucine, L-Leucine, Lysine base, L-Lysine Acetate, L-Phenylalanine, L-Threonine, L-Tryptophan, L-Valine, L-Histidine, L-Methionine. It was used mainly in patients with renal impairment since the lack of arginine in the mixture reduces protein kidney overload. Selective Amino Acid (SAA) formulae, in turn, which are rich in branched-chain amino acids, were used in patients with severe liver dysfunction that may lead to hepatic encephalopathy (1). The SAA admixture used by our pharmacy contains L-Alanine, L-Arginine, L-Cysteine hydrochloride monohydrate, Glycine, L-Isoleucine, L-Histidine, L-Leucine, L-Lisine acetate, L-Proline, L- Serine, L-Threonine, L-Tryptophan, L-Valine.

  1. Plauth M, Cabré E, Riggio O, Assis-Camilo M, Pirlich M, Kondrup J, et al. ESPEN Guidelines on Enteral Nutrition: Liver disease. Clin Nutr. 2006 Apr;25(2):285–94.

3.4- It is not clear what preparations of lipid emulsions were used. What is the relationship with the occurrence of cholestasis depending on the treatment, including lipid and glucose supply?

As reported in the revised version of the manuscript, two types of lipid admixtures were used at first: Lipofundin S® and Omegaven®. Lipofundin S® is an admixture containing soy oil, while Omegaven®  is a compound made of fish oil, egg phospholipids, and dl-alpha-tocopherol, usually indicated for adult patients. As previously reported in the literature, specific lipid emulsions with varied fatty acid composition, such as fish oil, may reduce complications associated with TPN lipid administration (1,2). Hence, while the Lipofundin S® was added to the PN bag of all patients to provide the daily lipid intake, the Omegaven® was only added to the bags of those patients who presented severe liver damage, like PNAC or who needed additional immune system stimulation (3,4). During the course of the years, the two compounds were progressively substituted by Smoflipid®, which contains refined soy oil, medium chain triglycerides, refined olive oil, and omega-3 enriched fish oil. This admixture could be nutritionally considered a combination of Lipofundin S® and Omegaven®, thus making it easier to manage and reach a balanced lipid intake. However, in some selected cases of massive hypertriglyceridemia, Smoflipid® has been replaced by Lipofundin S® to benefit from substituting olive oil with faster clearance oils – in this case, soy oil (5). The changes in the type of lipid used over the years could have mildly influenced the results on cholestasis, which was added in the text among the study's limitations.

We strongly felt that you raised a very important point regarding the influence of lipids and glucose supply.  For this reason, an additional round of analysis was carried out, in which also lipids and glucose were included. The difference in TPN-related cholestasis and hypertriglyceridemia was also confirmed after taking into account glucose and lipid supply (Table 3).

  1. 2. Guthrie G, Burrin Impact of Parenteral Lipid Emulsion Components on Cholestatic Liver Disease in Neonates. Nutrients. 2021 Feb 4;13(2):508.
  2. 3. Orso G, Mandato C, Veropalumbo C, Cecchi N, Garzi A, Vajro P. Pediatric parenteral nutrition-associated liver disease and cholestasis: Novel advances in pathomechanisms-based prevention and treatment. Dig Liver Dis. 2016 Mar;48(3):215–22.
  3. 4. Diamond IR, Sterescu A, Pencharz PB, Kim JH, Wales PW. Changing the Paradigm: Omegaven for the Treatment of Liver Failure in Pediatric Short Bowel Syndrome. J Pediatr Gastroenterol Nutr. 2009 Feb;48(2):209–15.
  4. 5. Allam-Ndoul B, Guénard F, Barbier O, Vohl MC. A Study of the Differential Effects of Eicosapentaenoic Acid (EPA) and Docosahexaenoic Acid (DHA) on Gene Expression Profiles of Stimulated Thp-1 Macrophages. Nutrients. 2017 Apr 25;9(5):424.
  5. 6. Mateu-de Antonio J, Florit-Sureda M. New Strategy to Reduce Hypertriglyceridemia During Parenteral Nutrition While Maintaining Energy Intake. J Parenter Enter Nutr. 2016 Jul;40(5):705–12.

3.5- The large age range of the subjects, 0-18 years, justifies providing information on the doses used in parenteral nutrition. Merely providing information about hyper or hypo - the result of the study of minerals, for example, is insufficient. 

Thank you for your comment. The definition of hypo- or hyper- in the study of blood minerals was made based on the minerals referral value for age, as reported by: Lega S, Minute M. “Prontuario Pediatrico”, ottava edizione. Medico e Bambino sas, Trieste, 2018. This was added in the text.  

3.6- It seems to me that the whole construction of the tables is difficult to interpret.

Thank you for your suggestion. In the revised version of the attached paper, we rearranged the Tables to facilitate the data interpretation. Descriptive analyses are kept in Table 1 and Table 2 to give the reader an idea of the characteristics of the study population. Regarding the multivariate analysis, statistically insignificant results have been moved to the supplementary section, and only the most relevant results are reported in Table 3.

3.7- The daily volume of TPN, and time of application of TPN should not be evaluated without reference to the weight and age of the subjects.

Undoubtedly the volume of TPN provided for each patient was estimated based on the height and weight of the children and their water requirement, which can vary according to each clinical picture. Unfortunately, data about the patient’s weight were incomplete. We included them in the revised version of the paper that we attached. Age, nonetheless, was available for every patient and was included in every multivariate regression model to take this effect into account.

3.8- As I understand it, since the nutritional mixtures were prepared in the hospital pharmacy individually for each patient, the requirement for personalized approval for pediatric patients was met, yet such a conclusion was made in the conclusions.

We believe that the understanding of sex differences is valuable knowledge per se and can improve the already personalized approach given to pediatric patients. This is true while preparing the TPN bag formula and as a clinical reminder when preventing TPN-associated complications. Anyway, we think that the relevance of this study resides in the availability of a large sample size to whom TPN is routinely administered in the oncologic setting. This allowed us to explore sex differences in TPN composition and complications, performing a gender analysis which is something unexplored in literature, especially in the pediatric field.

3.9- Since a single patient could be analyzed multiple times, were there differences in the studies between periods of nutritional treatment?

Thank you for the question. Unfortunately, this issue was not investigated. Each new TPN session was considered independent from the previous ones. This was included as a study limitation and can be a starting point for future research.

3.10- There are a lot of caveats that should be clarified if the study could be useful in clinical practice.

In the revised version of the paper, we tried to answer all the issues raised by the reviewers. Even with the limitations mentioned in the text, our article may be relevant since gender medicine is a very emerging field in pediatrics. Every new result is a single step towards specific recommendations about treatment and risk management, a single brick set on what we hope will be a wall of solid knowledge for sex-personalized care.

Round 2

Reviewer 1 Report

Thanks for the changes you did. Now I think the paper is better